# Health co-inquiry in migraine: Online participation and stakeholder experiences before and during the COVID-19 pandemic

**Camden L. Baucke[1], Lauren S. Seifert[2]\*, Kara Kaelber[2]**

**1** Counseling, Eastern Michigan University, Ypsilanti, MI, United States of America, **2** Counseling & Psychology Department, Malone University, Canton, OH, United States of America

\* LSEIFERT@malone.edu

## Abstract

A migraine is more than head pain, and chronic migraine can dramatically impact a person and those around her/him/them. To better understand those effects it is important to study the experiences of persons with migraine and their caregivers, family, friends, and health and mental health providers. When they collaborate, stakeholders may improve outcomes for persons with chronic migraine. One type of stakeholder cooperation is Health Co-Inquiry, involving a person-centered approach, activation of persons toward collaboration and improved health, evidence-based practice, and integrated care. The current study investigated Health Co-Inquiry at online forums, blogs, and bulletin boards where people came together to discuss migraine. A "Bifurcated Method" was used to conduct inductive, thematic analyses, quantitize themes, and cross-check themes using a robot program, which crawled the Internet to gather data about stakeholder sites and posts related to migraine. Key themes in the online narratives of migraine stakeholders included seeking and providing advice, help, and information. In addition, giving personal stories and testimonials, selling computer applications and products, and providing misinformation were frequent. Differences in the types of posts by various stakeholder groups were identified and may inform researchers about their varied perspectives and goals. Remarkably, migraine is still migraine–before a pandemic and during it. As such, migraineur concerns remained stable across thematic analyses of blog and forum posts before and during the worldwide COVID-19 pandemic.

**Data Availability Statement:** The data are in the Supplemental Information file. Data are also available at https://www.clovepress.com/downloads.html.

## Introduction

A migraine is much more than head pain; it can be accompanied by vertigo, nausea, vomiting, and other symptoms [1, 2]. Hypersensitivity to sensory stimuli in multiple domains (e.g., visual, auditory) is common, and it may persist during inter-ictal intervals, suggesting anomalies in multisensory integration [3]. The pain of migraine can be excruciating, and the level of disability may be high (e.g., moderate to severe) [4, 5]. Thus, the World Health Organization (WHO) has labeled migraine among the top 20 disabling conditions, acknowledging the strain

**Funding:** This study was supported by National Science Foundation CC-NIE Award Number #1541342 to our institution for technological development of research. We received a grant sub-award for the development of our project. The funders had no role in study design, data collection and analysis, decision to publish, or preparation of the manuscript.

**Competing interests:** No authors have competing interests.

that it places on caregivers, health and mental health providers, employers, governmental and non-governmental institutions, and migraineurs around the globe [6].

One approach to managing a chronic health condition like recurring migraine is Health Co-Inquiry, which is "a process whereby stakeholders in management of chronic health (or mental health) conditions collaborate, with efforts toward stakeholder activation, person-centeredness, evidence-based practice, and integrated care" [7, 8]. As stakeholders work together, they might better understand each other's perceptions and goals, thereby improving outcomes. In the new millennium, internet use in the general population has increased dramatically such that there are more than 1.9 billion websites and over 4 billion online users [9]. Along with this increase in internet use, people with chronic conditions might be seeking and/or sharing information, support, and services online (e.g., using WebMD™, blogging about their conditions). Therefore, it is reasonable to conjecture that stakeholders in chronic conditions may "health co-inquire" online.

The purpose of the current study was to investigate Health Co-Inquiry on the Internet among stakeholders in migraine (especially among those who have had more than one migraine). The current authors borrowed the Bifurcated Method [7] in order to implement an inductive thematic analysis [10] and to use quantitization via a robot computer program that counts the frequencies of specified words [7]. This study's fundamental goal was to identify key themes among the online artifacts that migraine stakeholders have posted on publicly available websites. The project began in 2017–2018, at a time that preceded the worldwide pandemic linked to SARS-CoV2 (COVID-19), but analyses continued as the pandemic happened; as such, this research followed the development of themes at migraine blogs and forums before and during the pandemic [14]. Ultimately, the project adds to an understanding of migraine stakeholders' experiences over time.

## Method

### Participants and safeguards

This study was reviewed and approved by the institutional review board (IRB) for human research at Malone University. No informed consent was required, because the study involves the use of publicly available narratives on the Internet. There were no participants; however, there were "subjects" in the sense that the persons posting at those sites were viewed as such, even though the current researchers had no interactions with them (with the researchers never posting on the URLs ["Uniform Resource Locators"]). Although data are archival, safeguards were put in place to help provide a buffer between this report and the users whose online narratives were studied. Before data collection, the institutional review board at the second/third authors' university approved the project protocol through an expedited review procedure. Second, no internet monikers of users are reported, and quotes have been redacted in order to decrease the chances that a reader might find them by searching on Google™ or a similar search engine. Third, a list of URLs is not in our manuscript but is available in the S1 File and at www.clovepress.com/downloads. Those safeguards are consistent with recommendations for internet research [11].

Participant characteristics from online posts are challenging to discern. It is relatively straightforward to distinguish stakeholder groups (i.e., from the forums on which they posted and the content of their messages, e.g., migraineurs, caregivers, professional health and mental health providers, organizations that provide care, non-profit organizations, government agencies, researchers). Yet, the typical types of demographic data that are available in survey research were not generally accessible from online blog/forum posts. As is mentioned below, this is a limitation of the study.

The most evident demographics (i.e., from a random sample of 20 websites from among our larger set of 92 URLS) were age and gender. Among persons with migraine, 30% (3/10) were male, 30% (3/10) were female, and 40% (4/10) had indiscernible gender based on the information they posted. From the same set of 20 URLs, for caregiver posts, 50% (1/2) appeared to be female, while the other blogger had indiscernible gender. Finally, among provider/other stakeholder narratives, 12.5% (1/8) was female, while the others were posts by agencies/groups that did not appear to have one featured provider who had discernible gender. No additional demographic information was available, except that some bloggers (i.e., 40% [8/20]) mentioned their ages or age group (e.g., "older" [Migraineur #1]; ". . .am 58. . ." [Migraineur #7]; a female of child-bearing age [Migraineur #2]). This leads to a conclusion that most (if not all) posts were made by adults.

## Materials

As noted above, data are archival records, i.e., online posts on publicly available blogs and forums. In addition, the present researchers used a WebCrawler that was previously developed by others [7]. Their guidebook is publicly available online, and they offer assistance to researchers who seek to use online data from blogs and forums [12].

## Procedures

As was mentioned, the protocol for this project involved an established "Bifurcated Method" which was adapted from previous research [7]. Appendix A in S1 File shows the basic steps in the Bifurcated Method (see S1 File). They include: (1) inductive, thematic (qualitative) analyses of 10 websites from a larger list of URLs found via searches of Google™, Ask™, Bing™, and Yahoo™, and (2) quantitizing of frequencies of keywords at URLs via a robot program [7]. Inclusion criteria were that a website must have had posts about the topic (i.e., migraine), must have included narrative of more than 40 characters, must have been in English, and must have been accessible without login/password. A master list of 92 viable URLs was created from internet searches using "migraine", "migraine blog", and "migraine forum" as search terms.

A random number generator was used to produce the list of 10 URLs, which were investigated (inductively) for themes in February 2018. As the COVID-19 pandemic happened, an additional thematic analysis was conducted with 20 randomly selected URLs in January/February 2020. Webcrawls occurred after inductive, thematic analyses (see below). Webcrawls with the robot program were performed periodically on 10 URLs (selected at random from the master list), or, alternatively, on as many URLs as the robot program could crawl (given internal and external constraints, like the computing power available on the project server and whether sites would allow crawling). The time course of the study was (1) inductive, thematic analysis [before COVID-19], (2) webcrawls [before COVID-19], (3) additional inductive, thematic analyses [during COVID-19], and (4) additional webcrawls [during COVID-19].

**Qualitative analyses.** Thematic, inductive analyses were performed by establishing frames of analysis, domains within frames, and themes within domains [10]. A fourth step was added that was not detailed by previous authors [7]. It included finding sub-themes within themes when possible. Frames of analysis are the individual pieces of information that are analyzed and compared. In the current study, a single post by a single user was the frame of analysis. Each post was read, analyzed, and compared to other posts by other users.

Domains are discernible categories within frames. For example, one might use subject characteristics as domains (e.g., male v. female; young adult v. older adult). Alternatively, one might use the views that subjects have expressed to categorize them into domains (e.g., how subjects have responded to screening questions or specific survey items). "Stakeholder group"

(i.e., migraineur, caregiver, provider [health or mental health], organization/agency, researcher, and employer [of the migraineur]) was the domain in this study (see the columns in S1 File: Appendix D).

Once frames of analysis and domains were identified, the authors read and scrutinized the narratives in order to find themes in them. This is an arduous process that is inductive, and the present researchers worked independently to remain open to what subjects had to say and to create lists of themes. After devising the lists, the investigators met for discussions about perceptions of themes. In order to be regarded as a key theme, a topic had to have been independently identified by at least 2 of the 3 authors (S1 File: see Appendix A, Steps 5 & 6). A significant amount of time was dedicated to achieving consensus—especially in meetings with the first two authors, who scoured their notes and those from the third author and discussed them [10]. Inter-rater agreement was calculated for inductive analyses during the COVID-19 pandemic (January/February-2020) but not for those in February-2018, which is a drawback in the method. More is mentioned about this in the Limitations section, below.

Saturation of themes in the inductive analysis was tabulated at two points during our project as a means of instilling additional rigor in the analyses (February 2018 and January/February 2020). During the initial phase of this study (2018–2019: prior to the COVID-19 pandemic) [14] the first and second authors divided up 64 URLs from the vetted list that was available from Seifert et al [7]. They each visited the first 10 posts/narratives at each website and evaluated the presence/absence of the themes from a list that was compiled by the three authors during their meetings regarding the inductive analysis. If a website did not have 10 posts, then the authors evaluated whatever narratives were present. This yielded saturation data from 41 URLs and a total of 386 posts. As will be discussed below, some analyses (e.g., January/February-2019; May-2021) included up to 92 URLs; as migraine URLs were taken down, new ones were posted on the Internet, and more URLs became crawlable with the robot program.

**Web crawl and quantitization of word frequencies.** As part of the Bifurcated Method, a robot program was used. It crawls the WWW and analyzes word frequencies at URLs that are specified by the researchers. As was mentioned above, search engines were used to find websites related to migraine. Only URLs meeting inclusion criteria (above) were entered into the web-crawling program. Generally, a crawl were conducted on as many of the vetted URLs as would permit crawling; occasionally, "snapshot" crawls were executed whereby a set of 10 randomly selected URLs was submitted to the crawler as a cross-checking mechanism; the shorter crawl was then compared to the most recent longer crawl(s).

The robot program conducted URL crawls (see crawlable URLS and data at www.clovepress.com/downloads). If it encountered a website that would not permit such a crawl, it yielded a total word frequency = 0 for the site. Such a result led the researchers to check the site to be sure the web address was correct and eliminate the URL from additional analyses if it did not seem crawlable. The total number of URLs that were crawled was influenced by a number of factors, such as whether a website was still active, permitted crawling, and/or had changed to require login for access. Furthermore, because the robot program used large amounts of random access memory (RAM), the researcher's institution's server sometimes "timed out" the program in order to provide RAM for other programs and processes. The foregoing factors account for the differences in total numbers of crawled URLs across attempts (in Appendices E and F in S1 File).

Part of the Bifurcated Method involves devising a general dictionary related to chronic health conditions; the current authors borrowed the general dictionary from originators of the Bifurcated Method [7]. However, an additional condition-specific dictionary for migraine was developed within the present research (S1 File: Appendix B). The current study began prior to

the US Food and Drug Administration (FDA) approvals for Aimovig$^{TM}$, Emgality$^{TM}$, and Ajovy$^{TM}$ which use monoclonal antibodies to bind calcitonin gene-related peptide (CGRP) receptors (Aimovig) or to bind the CGRP peptide itself (Emgality, Ajovy) [13, 17]. The proper names of those medications and other CGRP-related terms were not included in the 2018 migraine-specific dictionary but were added to the migraine-specific dictionary in 2019 webcrawls and those that followed (Appendices B, E, and F in S1 File).

The migraine-specific dictionary was compiled by consulting the second and third authors' university library. An online database was searched for the 10 most recent scholarly books about migraine for which tables of contents were accessible (i.e., EBSCO$^{TM}$, Ebsco Industries, Inc., n.d.; https://www.ebscoind.com/about-us/). The books' tables of contents yielded key terms related to migraine. Those terms are, presumably, used by experts in the field, since they appear in their books (see S1 File: Appendix B). Therefore, they were used in the migraine-specific dictionary for the present study. Inclusion criteria were that the term had to be related to migraine, health, mental health, and/or conditions that are comorbid with migraine. Nouns, verbs, adjectives, and adverbs were all permitted, and variations of words (like singular and plural forms) were frequently included in order to optimize the chance of finding nuances within the webcrawl data. Exclusion criteria were that a word did not satisfy inclusion criteria or that it was an article, preposition, or conjunction. The robot program provides a list of all dictionary (general and condition-specific) terms found at each website. It creates a spreadsheet showing comparative word frequencies for all URLs that were successfully searched. In addition, it supplies a list of the most frequent dictionary terms across all URLs.

**Comparing quantitized data to qualitative themes.** Following inductive thematic analysis and the web crawl, word frequencies from the crawl were scrutinized for relevance (or lack of relevance) to the themes from the inductive analysis. This is a critical step for instilling rigor in the Bifurcated Method, because one can compare the themes from the inductive analysis to the themes that seem to be represented by the most frequent terms (as counted by the robot program) at the URLs that were crawled [7].

**Follow-up to pre-pandemic analyses.** Although the initial inductive, thematic analysis was conducted in February-2018 (i.e., before the onset of the worldwide COVID-19 pandemic [14]), the researchers continued to review data in the three years that followed. This added rigor and enabled discernment about whether the COVID-19 pandemic, which began in Wuhan, China, in November-2019 [14], was appearing as a theme among posts. Keeping in mind that blogs and forums can change over time as new posts are made, it was important to evaluate whether the worldwide pandemic was a topic on migraine blogs and forums.

In January/February-2020, the first and second authors returned to migraine-related URLs in order to reevaluate themes and recalculate values for theme saturation. A random number generator was used to select 20 URLs from the master list that is mentioned above. In analyses during COVID-19, for practical purposes, the researchers collapsed stakeholder groups into three categories: migraineurs, caregivers, and providers/agencies/other. This was done, because migraine providers, agencies, and non-profit organizations seemed to post remarkably similar information. Each of the first two authors individually perused one post from each of the 20 URLs to determine: (1) Of what stakeholder group was the blogger/poster a member? (2) What themes were represented in the post?

## Results

### Themes in stakeholder narratives

Appendix D (S1 File) indicates themes in stakeholder narratives that were discerned via the inductive (qualitative) analysis in February-2018. As was mentioned, in pre-COVID-19

analyses, six stakeholder domains were identified: people with migraine, caregivers, health providers, researchers, organizations-agencies-government entities, and employers. Among the 10 URLs that were inductively reviewed by the present authors in 2018, there were 28 thematic narratives. Twenty of them provided information about migraine, but four (of the twenty) appeared to contain incorrect information (e.g., that a treatment is scientifically proven to cure migraines when it is not).

In 2018, the key theme among all narratives and all URLs was "testimonials and personal stories" that were provided by persons with migraine. Posts were replete with descriptions of pain and suffering. Researchers were unanimous in their agreement about this as a primary theme and that "pain and suffering" were prominent sub-themes. Saturation data indicate that personal stories/testimonials were present in 384/386 (99.48%) of posts and narratives. Among the personal narratives, an example is from Migraineur #1 who mentioned that he had to stop exercising "due to pain". Another person with migraine remarked about the experience of giving birth and that it was followed by "terrible migraines. . .[and] nausea" (Migraineur #2) and a third person listed his experience of migraine throughout his life, noting that the frequency of the attacks had progressively increased (Migraineur #7); moreover, he lamented the costs of medications, his despair at finding no effective treatments, and said that he was motivated "to live for [his] family".

A second theme among the posts was one of "providing support", with sub-themes related to "nurturing and caring for emotional and social needs". One-hundred-nine (of 386; 28.2%) were of this variety; most of the posts were by persons with migraine, although five providers and three caregivers made contributions. As an illustration, one post from an informal caregiver/friend describes her attempt to help her friend as they searched the Internet for information about treatments, such as botulinum toxin injections (Caregiver #3). Migraineur #5 provided some information about treatments that have worked for her and then stated "Good luck" in response to another blogger.

A third theme was one of "seeking information and/or help". Migraineurs and caregivers, seemed to search for help and answers about how to navigate life with migraine, and one employer posted. Researchers were in agreement about this as a fundamental theme. Saturation for the theme was 95/ 386 posts (24.61%). For instance, Migraineur #1stated that he was "interested in [others'] thoughts" about what had worked for them to alleviate headaches. Caregiver #2 mentioned that his wife had chronic headaches; he asked whether others "used Memantine" and what they had found in regard to using it for headaches.

A fourth theme related to "providing information" (73 of 386 posts; 18.9%). Such posts appeared to be focused on disseminating factual information. Provider #7 gave information about how to complete medical forms, and Provider #11 included contact information for call centers for suicide prevention and assistance with chemical dependency. Frequently, advice from persons with migraine was part of their personal stories. For instance, Migraineur #5 blogged about the treatments that had worked for her in response to another post with a request for information. Overall, providers and organizations (such as non-profit groups related to headaches and health management groups) tended to *focus* on providing information, as in Provider #3's free web course about managing migraines and Provider #2's links to organizations and resources for living with migraines.

A fifth, admittedly lesser theme among blogs, forums, and migraine websites was misinformation (38/386 sites; 9.8%). People are not perfect, and most persons with migraines are not trained medical professionals. Thus, it makes sense that there is incorrect information among migraine posts on the Internet. In general, misinformation seemed to be about how to use a particular medication or treatment method.

## Follow-up evaluation of themes in stakeholder groups

As noted, the first and second authors conducted additional thematic analyses in January/February 2020 during the COVID-19 pandemic [14] and for practical purposes, stakeholder groups were consolidated into: migraineurs, caregivers, and providers/agencies/other. With regard to determining the stakeholder group membership of a poster, inter-rater agreement was 100%, with 50% (10/20) being posts by migraineurs, 10% (2/20) as posts by caregivers, and 40% (8/20) being posts by providers/agencies/other.

In the follow-up analysis of themes, inter-rater agreement was 100% for "seeking information/help" and for "giving misinformation"; it was 90% for "providing support". However, inter-rater agreement was lower (80%) for "providing information" and (75%) "giving testimony/personal stories". The second author was using a more liberal criterion for identifying the latter themes in posts, and this led the researchers to use the first author's data, which were more conservative in the attribution of themes to posts and blogs.

From the beginning of 2018 through May-2021—before and during the worldwide pandemic from COVID-19 [14]—themes and stakeholder groups remained relatively stable at migraine forums and blogs. In January/February 2020, as fears grew globally about the spread of the coronavirus, key themes appeared to stay the same, but comparative saturation of themes may have changed, with "providing information" appearing as the most prominent theme (9/20; 45%), and "seeking information or help" being the second-most (8/20; 40%). "Personal stories and testimonials" were still present (5/20; 25%), but "providing support" seemed less frequent (1/20; 5%). Misinformation was also noted less often (1/20; 5%).

## Web-crawl data

As noted in the Method section, webcrawls were performed in order to provide quantitization of key words (related to chronic health conditions and to migraine). Appendices E and F in S1 File show crawl data from before the worldwide COVID-19 pandemic and during it. The initial webcrawl was conducted in January-2018, but we did not receive or review the web-crawl data until after our February-2018 thematic analyses were completed. The most recent crawl was executed in May-2021, when many regions of the U.S. were lifting restrictions on sizes of social gatherings and easing requirements for personal protective equipment (e.g., face masks) [15, 16].

Word frequencies were remarkably similar over time and across seven crawls in three years; so the researchers cross-checked URLs to be sure that they were still active. Most were, and slight changes in crawl word counts and the ordering of the highest frequency terms across sites indicates that some URLs were posting new information and experiencing new blog posts. "Migraine" was the highest frequency term in all crawls, and "ache" and "Ill" vied for second place among the top-ten most frequent terms (with the latter being specified to begin with a capital letter in order to denote its place at the start of an utterance). Additional high-frequency terms were: "pain", "work", "son", "doctor", "suffer", "trigger", "treatment", "symptom", "triptan", "medicine", "drug", "evaluation", and "gene". While COVID-19 was not specifically mentioned in later crawls, "calcitonin" and "gene-related" did appear in blogs, indicative of migraine stakeholder concerns about the FDA approval of an entirely new class of medications to treat migraines (see Appendices E and F in S1 File) [17].

## Cross-checking themes with crawl data

Crawl data are consistent with inductively-generated themes, being words that represent personal stories (e.g., of illness, pain, aches), providing support (e.g., family labels such as son, mother, father), seeking information and help or providing those (e.g., about doctors,

medications/drugs that work), or giving misinformation (e.g., naming a treatment that has not been scientifically evaluated for treatment of migraine). Webcrawl data appear to support researcher-generated themes and serve as a cross-checking mechanism to add rigor to qualitative analyses. Overall, word counts indicate that suffering and pain linked to migraine are profound and that migraine stakeholders are engaged in seeking and providing information and help/support online.

## Limitations

The current study has a number of drawbacks. First, as was mentioned, it is challenging to discern demographic information about migraine stakeholders from their online posts. This limits researchers' abilities to define the sample and frame the data. Given that more persons with migraine are female and that males with migraine may be under-diagnosed [18], it is concerning that we could not assess the role of gender in Health Co-Inquiry online. A second disadvantage of the present study was that not all URLs in the master list of 92 were crawled/crawlable at each occasion when the robot program was used; this was due to RAM limits and to sites that blocked crawling or that required user login. As such, crawl data may not provide a complete picture of migraine URLs and their narratives. A third drawback was that inter-rater agreement was calculated for the second thematic analysis but not the first; however, the researchers did seek consensus in the initial inductive analysis, which should have bolstered its rigor.

## Discussion

Overall, the Bifurcated Method [7] that was adapted here for the study of migraine, indicates that migraine is a significant challenge for people who experience it. Multiple stakeholder groups are active online in "Health Co-Inquiry" [7] as they share information and support/help for persons with migraine. Pain/suffering and nurturing/caring are key sub-themes in stakeholder online posts, and word counts from webcrawls support the current authors' conclusions about themes in narratives at migraine websites. Furthermore, our pre-COVID-19 January-2018 data are consistent with those reported by Seifert et al. in 2019 (see S1 File: Appendix F) with "migraine", "pain", "ache", and "ill" as the highest frequency terms in online posts [7].

In an online survey of migraineurs in Kuwait during the COVID-19 pandemic, more than 30% of patients sought relief outside of a physician's care (e.g., self-care, "traditional medicine" like Ayurveda, head-banding, blood cupping) [19], and although the researchers did not specifically ask about it, this might have included searching for help online. Moreover, in a study of Google[TM] Analytic (GA) data from a Spanish URL that provides information about migraine diagnosis and treatment, investigators reported increasing use of the website from 2015 to 2020, with the most frequent keyword searches being "days", "migraine", "pain", and "aura" [20]. Two of those keywords are among the highest frequency terms in all of the webcrawl searches that we have reported here (Appendices E and F in S1 File), and although the authors of the Spanish study [20] evaluated data from keyword searches (rather than forum posts), their word cloud is remarkably similar to our word clouds for narrative content at migraine websites and forums (with our word clouds available at www.clovepress.com/downloads). Together with the current data, the foregoing studies indicate that persons with migraine and other migraine stakeholders may be turning to Health Co-Inquiry collaborations, including those online, as they seek and offer help, support, and services related to migraine.

Regarding misinformation online, our data suggest that it is a challenge. As was mentioned above, we observed that incorrect information was posted at 38 of 386 sites (9.8%). Lavorgna et al. [21, 22] have studied incidents of "fake news" on a social media site for persons with Multiple Sclerosis (MS) and noted that 72 of 380 posts (18.95%) contained misinformation. While the percentage they reported is twice what we documented, this may be due to differences in the timing of the studies (theirs in 2015; ours from 2018–2021) and to their focus on one social media platform which is dedicated to MS [21, 22], while our research evaluated posts across many websites about migraine. In a different investigation about the spread of misinformation on Twitter during the Zika epidemic, valid facts prompted more retweets and attracted more users than fake news, overall; however, the latter were more viral and seemed to involve greater network diameter (the social distance between two users on a social media platform) [23]. Taken together, investigations of misinformation on health forums and social media indicate that there is a need for experts to contribute to online discussions about health so that persons in the general population who are seeking information on the Internet can be guided toward credible posts and away from misinformation [21].

The stress created by a chronic health condition can be immense, and stress associated with migraine is well-documented [2, 4, 5, 8]. It stands to reason that stressors apart from migraine may detract from one's ability to mobilize resources to cope with migraine [2, 6]. Although COVID-19 was not a fundamental topic of discussion at the migraine sites that we evaluated, this does not mean that the stress of the COVID pandemic has not impacted migraineurs. Self-reports from persons with migraine and their caregivers indicate that migraine contributes to stress and is influenced by it, thereby diminishing quality of life (QOL) [4, 5, 6, 8]. We believe that this is a topic for additional research, because stress linked to the COVID-19 pandemic is also evident [24], and it is unclear how things like COVID-19 infection, quarantines, lockdowns, and social distancing may have affected migraine and those who experience it.

We noted that COVID-19 did not loom large among topics at migraine forums during later crawls (S1 File: Appendix F). Perhaps, those suffering the effects of both migraine and COVID-19 were posting elsewhere (e.g., sites dedicated to COVID-19) [25]. On the other hand, as a topic directly linked to migraine relief, calcitonin gene-related peptide did begin to appear at URLs about migraine during our later crawls. Given the profound impact of migraine on the daily lives of people around the globe [2], it is not surprising that the concerns of those who post at migraine-related URLs appear to have continued to be on migraine, even during the COVID-19 pandemic. It seems as if *migraine remains migraine* as a source of suffering and pain. . .whether outside of the COVID-19 pandemic or within it.

## Supporting information

**S1 File. Appendices and data.**
(PDF)

## Author Contributions

**Conceptualization:** Camden L. Baucke, Lauren S. Seifert, Kara Kaelber.

**Data curation:** Camden L. Baucke, Lauren S. Seifert, Kara Kaelber.

**Formal analysis:** Camden L. Baucke, Lauren S. Seifert, Kara Kaelber.

**Funding acquisition:** Lauren S. Seifert, Kara Kaelber.

**Investigation:** Camden L. Baucke, Lauren S. Seifert, Kara Kaelber.

**Methodology:** Camden L. Baucke, Lauren S. Seifert, Kara Kaelber.

**Project administration:** Lauren S. Seifert.

**Resources:** Camden L. Baucke, Lauren S. Seifert, Kara Kaelber.

**Software:** Lauren S. Seifert, Kara Kaelber.

**Supervision:** Lauren S. Seifert, Kara Kaelber.

**Validation:** Camden L. Baucke, Lauren S. Seifert, Kara Kaelber.

**Visualization:** Camden L. Baucke, Lauren S. Seifert, Kara Kaelber.

**Writing – original draft:** Camden L. Baucke.

**Writing – review & editing:** Camden L. Baucke, Lauren S. Seifert, Kara Kaelber.

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
