## [Decision Letter · Decision Letter 0]

2 Sep 2021

PONE-D-21-19850

Health Co-Inquiry in Migraine: Online Participation and Stakeholder Experiences Before and During the COVID-19 Pandemic

PLOS ONE

Dear Dr. Seifert,

Thank you for submitting your manuscript to PLOS ONE. After careful consideration, we feel that it has merit but does not fully meet PLOS ONE’s publication criteria as it currently stands. Therefore, we invite you to submit a revised version of the manuscript that addresses the points raised during the review process.

Please submit your revised manuscript Sept 15Th If you will need more time than this to complete your revisions, please reply to this message or contact the journal office at plosone@plos.org. Please include the following items when submitting your revised manuscript:

We look forward to receiving your revised manuscript.

Kind regards,

Luigi Lavorgna

Academic Editor

PLOS ONE

Journal Requirements:

2. In line with PLOS' guidelines on detailed reporting (https://journals.plos.org/plosone/s/criteria-for-publication#loc-3), please ensure that you have provided sufficient detail in the Methods section regarding which blogs/websites/forums were selected for data extraction. If materials, methods, and protocols are well established, authors may cite articles where those protocols are described in detail, but your submission should include sufficient information to be understood independent of these references (https://journals.plos.org/plosone/s/submission-guidelines#loc-materials-and-methods). 

This study was supported by National Science Foundation  CC-NIE Award Number #1541342 to our institution for technological development of research. We received a grant sub-award for the development of our project.

Reviewers' comments:

Reviewer's Responses to Questions

**Comments to the Author**

1. Is the manuscript technically sound, and do the data support the conclusions?

Reviewer #1: Yes

Reviewer #2: Yes

2. Has the statistical analysis been performed appropriately and rigorously? 

Reviewer #1: I Don't Know

Reviewer #2: Yes

3. Have the authors made all data underlying the findings in their manuscript fully available?

Reviewer #1: Yes

Reviewer #2: Yes

4. Is the manuscript presented in an intelligible fashion and written in standard English?

Reviewer #1: Yes

Reviewer #2: Yes

5. Review Comments to the Author

Reviewer #1: The authors investigated Health Co-Inquiry at online forums, blogs, and bulletin boards where people came together to discuss migraine. Very interesting topic. The manuscript requires minor revision.

The authors should provide in the Introduction section a clearer definition of Health Co-Inquiry. The one provided (page 3, line 57): “which promotes patient (and other stakeholder) activation, person-centeredness, evidence-based practice, and integrated care.” not is enough.

As mentioned in the introduction section (page 3, line 59): “in the new millennium, internet use in the general population has increased dramatically”. However, the authors should also discuss that with rapid growth of Internet use by the general population in the new millennium, persons with chronic conditions and their caregivers may be using the Internet more in order to seek information, share advice, and give or find support and services… and “Therefore, it is reasonable to conjecture that stakeholders in chronic conditions may “health co-inquire” online.”

The discussion section needs to be expanded. The authors could discuss previous studies that applied the same approach and compare the results of these studies with theirs. Indeed, the main goal of this study, as stated by the authors, was to identify key themes among the online artifacts that migraine stakeholders have posted on publicly available websites. It would be interesting a comparison with the results of similar studies (not necessarily concerning migraine) (i.e., Seifert et al. Curr Psychol 2019).

Furthermore, the authors could highlight and examine in depth the topic of misinformation spread on the internet and how healthcare providers could face it (Lavorgna L et al. Mult Scler Relat Disord. 2018; Lavorgna L, et al. Interact J Med Res. 2017).

Reviewer #2: Nice study to evaluate experiences of persons with migraine and their caregivers, family, friends, and health and mental health providers before and during the COVID-19 Pandemic. Considering the female prevalence and attitude, 40% with indiscernible gender colud be a problem to better discuss in study limitations. Qualitative analysis is well conducted and data might be shown with table for quotes from patients and health care providers and word cloud. I suggest the authors to add some considerations or comments in the discussion paragraph to underlie the core results, wondering respect to general data arising during covid pandemic on stressfull and disabling symptoms and their impact on quality of life of patients.

6. PLOS authors have the option to publish the peer review history of their article (what does this mean?). If published, this will include your full peer review and any attached files.

Reviewer #1: No

Reviewer #2: No

---

## [Author Response · Author response to Decision Letter 0]

21 Oct 2021

RE: PLOS ONE manuscript # PONE-D-21-19850

Date: October 17, 2021

Dear Dr. Lavorgna,

Many thanks for your message letting us know that revisions were required for our paper about “Health Co-Inquiry in Migraine: Online Participation and Stakeholder Experiences Before and During the COVID-19 Pandemic”.

We have revised the manuscript per yours and the Reviewers’ comments.

Each comment is listed and addressed.

Many thanks from the Authors

****

Editor Requests:

An unmarked version of your revised paper without tracked changes. You should upload this as a separate file labeled 'Manuscript'

Author Response:

All of the above-required items have been included.

Editor Requests and Author Response:

You have suggested that we upload detailed lab protocols, but we do not have lab protocols, because this was a study of webforums and blogs. So, we have not uploaded lab protocols.

Editor Requests:

Make changes to financial disclosures if needed.

Author Response:

About financial disclosures, as you have recommended, the following is true. Please, add this statement to our Role of Funder statement:

****

Journal Requirements:

Author Response:

We have reviewed the requirements and have named the files accordingly.

2. In line with PLOS' guidelines on detailed reporting (https://journals.plos.org/plosone/s/criteria-for-publication#loc-3), please ensure that you have provided sufficient detail in the Methods section regarding which blogs/websites/forums were selected for data extraction. If materials, methods, and protocols are well established, authors may cite articles where those protocols are described in detail, but your submission should include sufficient information to be understood independent of these references (https://journals.plos.org/plosone/s/submission-guidelines#loc-materials-and-methods). 

Author Response:

We have posted our data online and have included all URLs for all sites that were crawled by our web-crawling application at:

https://www.clovepress.com/downloads.html

This is now mentioned in our manuscript in the Methods and the Results sections (pp. 4-5).

We note here, that we recently became aware that our robot program conducted a January-2018 webcrawl. We are not surprised about this, as our programmer was working on the program and conducting crawls during that time. So, we have replaced our February-2018 crawl data (Appendix E) with the January-2018 crawl data, in the interest of providing the earliest possible crawl data for comparison to later crawl data. 

This study was supported by National Science Foundation CC-NIE Award Number #1541342 to our institution for technological development of research. We received a grant sub-award for the development of our project.

Author Response:

Thank you. As we mentioned above, we request that you add the following to our manuscript financial disclosures:

Author Response:

Many thanks. There are no data restrictions and as we have mentioned above and in the manuscript, the data have been posted at:

https://www.clovepress.com/downloads.html

5. Please include your full ethics statement in the ‘Methods’ section of your manuscript file. In your statement, please include the full name of the IRB or ethics committee who approved or waived your study, as well as whether or not you obtained informed written or verbal consent. If consent was waived for your study, please include this information in your statement as well

Author Response:

The following has been added to our Method section (p. 4).

“This study was reviewed and approved by the institutional review board for human research at Malone University. No informed consent was required, because the study involves the use of publicly available narratives on the Internet.”

Author Response:

We have found no retracted manuscripts among our references.

Changes to our manuscript include reordering the former References: 7, 8, 9 because of text changes within the manuscript which warranted the renumbering. In addition, as part of our responses to Reviewers, we have added References 19 through 25 and cite them in our revised paper.

****

Reviewer Remarks:

Reviewer #1: The authors investigated Health Co-Inquiry at online forums, blogs, and bulletin boards where people came together to discuss migraine. Very interesting topic. The manuscript requires minor revision.

The authors should provide in the Introduction section a clearer definition of Health Co-Inquiry. The one provided (page 3, line 57): “which promotes patient (and other stakeholder) activation, person-centeredness, evidence-based practice, and integrated care.” not is enough.

Author Response:

Many thanks for this critique. We have now provided the definition of Health Co-Inquiry as it was given by previous authors. It is quoted on p. 3 with source information:

One approach to managing a chronic health condition like recurring migraine is Health Co-Inquiry, which is “a process whereby stakeholders in management of chronic health (or mental health) conditions collaborate, with efforts toward stakeholder activation, person-centeredness, evidence-based practice, and integrated care.”

Reviewer #1 Remark:

As mentioned in the introduction section (page 3, line 59): “in the new millennium, internet use in the general population has increased dramatically”. However, the authors should also discuss that with rapid growth of Internet use by the general population in the new millennium, persons with chronic conditions and their caregivers may be using the Internet more in order to seek information, share advice, and give or find support and services… and “Therefore, it is reasonable to conjecture that stakeholders in chronic conditions may “health co-inquire” online.”

Author Response:

Many thanks for noting the need for a logical connection between growth in internet use and Health Co-Inquiry. The following sentence has now been added on p. 3:

“Along with this increase in internet use, people with chronic conditions might be seeking and/or sharing information, support, and services online (e.g., using WebMDTM, blogging about their conditions).”

Reviewer #1 Remark:

The discussion section needs to be expanded. The authors could discuss previous studies that applied the same approach and compare the results of these studies with theirs. Indeed, the main goal of this study, as stated by the authors, was to identify key themes among the online artifacts that migraine stakeholders have posted on publicly available websites. It would be interesting a comparison with the results of similar studies (not necessarily concerning migraine) (i.e., Seifert et al. Curr Psychol 2019).

Author Response:

Many thanks. Accordingly, we have added the following to our Discussion on p. 17:

Furthermore, our pre-COVID-19 January-2018 data are consistent with those reported by Seifert et al. in 2019 (see Appendix F) with “migraine”, “pain”, “ache”, and “ill” as the highest frequency terms in online posts.7 

In an online survey of migraineurs in Kuwait during the COVID-19 pandemic, more than 30% of patients sought relief outside of a physician’s care (e.g., self-care, “traditional medicine” like Ayurveda, head-banding, blood cupping).19, and although the researchers did not specifically ask about it, this might have included searching for help online. Moreover, in a study of GoogleTM Analytic (GA) data from a Spanish URL that provides information about diagnosis and treatment, investigators reported increasing use of the website from 2015 to 2020, with the most frequent keyword searches being “days”, “migraine”, “pain”, and “aura”.20 Two of those keywords are among the highest frequency terms in all of the web-crawl searches that we have reported here (Appendices E and F), and although the authors of the Spanish study20 evaluated data from keyword searches (rather than forum posts), their word cloud is remarkably similar to our word clouds for narrative content at migraine websites and forums (with our word clouds available at www.clovepress.com/downloads). Together with the current data, the foregoing studies indicate that persons with migraine and other migraine stakeholders may be turning to Health Co-Inquiry collaborations, including those online, as they seek and offer help, support, and services related to migraine. 

Reviewer #1 Remark:

Furthermore, the authors could highlight and examine in depth the topic of misinformation spread on the internet and how healthcare providers could face it (Lavorgna L et al. Mult Scler Relat Disord. 2018; Lavorgna L, et al. Interact J Med Res. 2017).

Author Response:

Thank you. We agree that misinformation is a concern. We have added Discussion about this on pp. 17-18:

Regarding misinformation online, our data suggest that it is a challenge. As was mentioned above, we observed that incorrect information was posted at 38 of 386 sites (9.8%). Lavorgna et al.21,22 have studied incidents of “fake news” on a social media site for persons with Multiple Sclerosis (MS) and noted that 72 of 380 posts (18.95%) contained misinformation. While the percentage they reported is twice what we documented, this may be due to differences in the timing of the studies (theirs in 2015; ours from 2018-2021) and to their focus on one social media platform which is dedicated to MS21,22, while our research evaluated posts across many websites about migraine. In a different investigation about the spread of misinformation on Twitter during the Zika epidemic, valid facts prompted more retweets and attracted more users than fake news, overall; however, the latter were more viral and seemed to involve greater network diameter (the social distance between two users on a social media platform).23 Taken together, investigations of misinformation on health forums and social media indicate that there is a need for experts to contribute to online discussions about health so that persons in the general population who are seeking information on the Internet can be guided toward credible posts and away from misinformation.21

Reviewer #2 Remark: 

Nice study to evaluate experiences of persons with migraine and their caregivers, family, friends, and health and mental health providers before and during the COVID-19 Pandemic. 

Author Response:

Many thanks.

Reviewer #2 Remark:

Considering the female prevalence and attitude, 40% with indiscernible gender colud be a problem to better discuss in study limitations. 

Author Response:

Thank you. We have added the following information to our Limitations section on p. 16:

“Given that more persons with migraine are female and that males with migraine may be under-diagnosed24, it is concerning that we could not assess the role of gender in Health Co-Inquiry online.”

Reviewer #2 Remark:

Qualitative analysis is well conducted and data might be shown with table for quotes from patients and health care providers and word cloud. 

Author Response:

Many thanks for your suggestions. We have included examples in the text of the paper regarding online posts. For example, on p. 12, we wrote:

Among the personal narratives, an example is from Migraineur #1 who mentioned that he had to stop exercising “due to pain”. Another person with migraine remarked about the experience of giving birth and that it was followed by “terrible migraines…[and] nausea” (Migraineur #2) and a third person listed his experience of migraine throughout his life, noting that the frequency of the attacks had progressively increased (Migraineur #7); moreover, he lamented the costs of medications, his despair at finding no effective treatments, and said that he was motivated “to live for [his] family”.

And on p. 13, we stated that:

For instance, Migraineur #1stated that he was “interested in [others’] thoughts” about what had worked for them to alleviate headaches. Caregiver #2 mentioned that his wife had chronic headaches; he asked whether others “used Memantine” and what they had found in regard to using it for headaches.

Author Response continued:

We note, here, that we did not make a table of quotes, as we believe that it might be slightly redundant with the examples in the body of our paper. However, we are willing to revisit this issue if Reviewers believe that a table is needed in addition to the text in our paper about post contents.

Author response continued:

About the mention of word clouds, we greatly valued this suggestion and have now added the following to p. 17. It is within the context of the paragraph that we added in response to Reviewer #1 comments (above), as well.

“…and although the authors of the Spanish study20 evaluated data from keyword searches (rather than forum posts), their word cloud is remarkably similar to our word clouds for narrative content at migraine websites and forums (with our word clouds available at www.clovepress.com/downloads).”

Reviewer #2 Remarks:

I suggest the authors to add some considerations or comments in the discussion paragraph to underlie the core results, wondering respect to general data arising during covid pandemic on stressfull and disabling symptoms and their impact on quality of life of patients.

Author Response:

Many thanks. We have added the following to our Discussion on p. 18:

The stress created by a chronic health condition can be immense, and stress associated with migraine is well-documented.2,4,5,8 It stands to reason that stressors apart from migraine may detract from one’s ability to mobilize resources to cope with migraine.2,6 Although COVID-19 was not a fundamental topic of discussion at the migraine sites that we evaluated, this does not mean that the stress of the COVID pandemic has not impacted migraineurs. Self-reports from persons with migraine and their caregivers indicate that migraine contributes to stress and is influenced by it, thereby diminishing quality of life (QOL).4,5,6,8 We believe that this is a topic for additional research, because stress linked to the COVID-19 pandemic is also evident25, and it is unclear how things like COVID-19 infection, quarantines, lockdowns, and social distancing may have affected migraine and those who experience it.

Author Closing Remark:

Many thanks to the Editor and Reviewers for the tremendous care and attention paid to our manuscript. We are grateful for your remarks and hope that our revisions are appropriate.

With best regards,

The Authors

---

## [Editor Report · Decision Letter 1]

9 Nov 2021

Health Co-Inquiry in Migraine: Online Participation and Stakeholder Experiences Before and During the COVID-19 Pandemic

PONE-D-21-19850R1

We’re pleased to inform you that your manuscript has been judged scientifically suitable for publication and will be formally accepted for publication once it meets all outstanding technical requirements.

Kind regards,

Luigi Lavorgna

Academic Editor

PLOS ONE
---

## [Editor Report · Acceptance letter]

17 Nov 2021

PONE-D-21-19850R1 

Health Co-Inquiry in Migraine: Online Participation and Stakeholder Experiences Before and During the COVID-19 Pandemic 

Dear Dr. Seifert:

I'm pleased to inform you that your manuscript has been deemed suitable for publication in PLOS ONE. Congratulations! Your manuscript is now with our production department. 

Kind regards, 

on behalf of

Dr. Luigi Lavorgna 

Academic Editor

PLOS ONE